# The Effect of Exercise and Nutritional Interventions on Body Composition in Patients with Advanced or Metastatic Cancer: A Systematic Review

**DOI:** 10.3390/nu14102110

**Published:** 2022-05-18

**Authors:** Oscar Barnes, Rebekah L. Wilson, Paola Gonzalo-Encabo, Dong-Woo Kang, Cami N. Christopher, Thomas Bentley, Christina M. Dieli-Conwright

**Affiliations:** 1Green Templeton College, University of Oxford, Oxford OX2 6HG, UK; oscar.barnes@gtc.ox.ac.uk (O.B.); thomas.bentley@gtc.ox.ac.uk (T.B.); 2Division of Population Sciences, Department of Medical Oncology, Dana-Farber Cancer Institute, 375 Longwood Avenue, Boston, MA 02215, USA; rebekahl_wilson@dfci.harvard.edu (R.L.W.); paola_gonzaloencabo@dfci.harvard.edu (P.G.-E.); dong-woo_kang@dfci.harvard.edu (D.-W.K.); cameron_christopher@dfci.harvard.edu (C.N.C.); 3Department of Medicine, Harvard Medical School, Boston, MA 02215, USA; 4Department of Epidemiology, Boston University, Boston, MA 02118, USA

**Keywords:** cancer, metastatic, advanced, exercise, nutrition, lean muscle mass, fat mass, body composition

## Abstract

Advanced and metastatic cancers significantly alter body composition, leading to decreased lean mass and variable effects on fat mass. These effects on body composition are associated with significant physical dysfunction and poor prognosis in patients with cancer. Whilst exercise and nutritional interventions are likely to be of benefit in counteracting these effects, relatively little is known about using such interventions in patients with advanced or metastatic cancer. Therefore, in this systematic review we examine the effect of exercise and combined exercise and nutritional interventions on lean mass and fat mass among patients diagnosed with advanced or metastatic cancer. Following PRISMA guidelines, we identified 20 articles from PubMed, EMBASE, CINAHL, Cochrane CENTRAL, PEDro, SPORTDiscus, and REHABDATA. Overall, advanced or metastatic cancer populations comprising of mixed cancer types were most commonly examined (*n* = 8) with exercise or combined exercise and nutritional interventions being well-tolerated with few adverse effects. Both intervention approaches may preserve lean mass, while only combined interventions may lead to alterations in fat mass. However, further exercise and nutritional studies are needed to definitively understand their effects on body composition. As exercise and nutrition-related research continues in this understudied population, the knowledge gained will help guide supportive clinical treatments.

## 1. Introduction

Patients with advanced and metastatic cancer are now living longer due to advancements in cancer therapy [1,2,3]. Nevertheless, many patients experience substantial physiological and psychological disruption, including profound changes in their body composition. Some cancers, namely breast [4] and prostate [5], are associated with significant weight gain, whilst others, such as gastrointestinal cancer, lead to significant weight loss [6]. Such alterations may be mediated by inadequate dietary intake and reduced physical activity, alongside the catabolic effects of the underlying malignant disease or cancer treatment [7,8]. Body composition changes in patients with advanced or metastatic cancer are associated with significant impairments in physical function, increased psychological burden [9,10,11], decreased quality of life [11,12], and increased fatigue [13,14].

One major change is loss of lean mass, mainly through skeletal muscle loss. Low lean mass has emerged as a driver of physical dysfunction and poor quality of life [15], and also as a poor prognostic feature in patients with advanced cancer [16,17,18], associated with increased mortality [17] and greater treatment toxicity [19,20]. The consequences of low lean mass extend beyond the individual patient, leading to longer hospital stays and greater hospital-related costs [21,22]. Importantly, loss of lean mass often occurs as a part of an established syndrome, namely cachexia and sarcopenia, both prevalent conditions in patients with advanced or metastatic cancer [23,24]. Cachexia, a tissue wasting syndrome characterized by weight loss, mainly from skeletal muscle loss with or without fat loss, is frequently reported in over 50% of patients with cancer [25,26] and is noted to reduce survival [27]. Sarcopenia is a separate syndrome characterized by loss of skeletal muscle alongside physical dysfunction [28]. Whilst often associated with aging [29], sarcopenia commonly occurs in patients with advanced or metastatic cancer [30,31].

In parallel with loss of lean mass, patients with advanced or metastatic cancer experience significant fat mass changes [32]. Fat mass can either be gained [33] or lost [34] in patients with advanced or metastatic disease depending on cancer type, stage, and treatments. For example, loss of fat mass is particularly prevalent in gastrointestinal cancers [35], whereas patients with prostate cancer undergoing androgen deprivation therapy have been shown to gain fat mass whilst losing lean mass across the first few months of treatment [5]. Alterations in fat mass significantly impact the metabolism [36], leading to conditions such as insulin resistance [37] and chronic systemic inflammation [38]. Furthermore, obesity is highly prevalent among people diagnosed with cancer [39], acting as a key risk factor [40] and predictor of metastases, particularly among breast cancer survivors [41]. Importantly, simultaneous differences in lean mass and fat mass relative to healthy patients are prevalent, such as in sarcopenic obesity [18].

Exercise has emerged as an effective therapeutic adjunct to conventional cancer therapies and has been shown to improve the quality of life and physical function of patients with cancer [42]. However, patients with metastatic or advanced cancer remain largely understudied compared to patients at other stages [43], although emerging evidence suggests exercise to be safe for this population [44,45]. Despite the likely benefits of exercise in this population, changes to body composition also depend on nutritional status. Consequently, the addition of a nutritional component is often encouraged [46]; however, the effect of adding a nutritional component to exercise, compared to exercise alone, has not been fully examined among patients with advanced or metastatic cancers. The mechanisms through which exercise may target cancer-related changes in body composition have been reviewed elsewhere [6], and include targeting systemic inflammation, hypogonadism, protein synthesis, and oxidative metabolism. It has also been tentatively suggested that exercise may help to counter malignant disease itself [47,48]. The evidence supporting nutritional intervention [49,50] alongside appropriate exercise as key components of supportive care in advanced cancer is strongly reflected in recent clinical practice guidelines. The latest European Society for Medical Oncology (ESMO) clinical practice guideline for cancer-associated cachexia recommends multimodal supportive management that includes both nutritional support and muscle training where appropriate to provide anabolic stimulus [51]. Corresponding guidelines from the American Society of Clinical Oncology (ASCO) and the European Society for Clinical Nutrition and Metabolism (ESPEN) emphasize the need to initiate care that includes nutritional support as soon as a patient is diagnosed with advanced cancer [52,53,54].

Despite the heightened interest in exercise and nutritional interventions in patients with more advanced disease, the evidence base is not clearly established. Previous reviews have provided preliminary evidence that exercise can improve quality of life and physical function in patients with advanced cancer [55,56]. Additionally, some reviews have included body composition [44,57,58,59,60], suggesting that exercise can significantly increase lean mass [61,62,63] and alter fat mass [57]. However, no reviews have summarized the effects of combined exercise and nutritional interventions in patients with advanced or metastatic cancer, particularly with regards to their effects on overall body composition. Therefore, this review examines the effect of exercise and combined exercise and nutritional interventions on lean mass and fat mass among cancer patients diagnosed with advanced or metastatic cancer. We hypothesize that exercise or combined exercise and nutritional interventions will maintain or improve body composition compared to controls in patients with advanced or metastatic cancer.

## 2. Materials and Methods

This systematic review was conducted according to the Preferred Reporting Items for Systematic Reviews (PRISMA) guidelines [64]. It is registered under PROSPERO ID CRD42022314284.

### 2.1. Search Strategy

The search was performed in PubMed, Excerpta Medica Database (EMBASE), Cumulative Index to Nursing and Allied Health Literature (CINAHL), Cochrane Central Register of Controlled Trials (CENTRAL), Physiotherapy Evidence Database (PEDro), SPORTDiscus, and National Rehabilitation Information Center Database (REHABDATA). Keywords were developed by initial idea gathering and searching a thesaurus for the following concepts: advanced/metastatic, malignancy/cancer, exercise/physical activity, and body composition. Further key words were then added through review of the titles and abstracts of relevant papers. For each database, specific medical subject headings were then added to the search terms. Databases were last searched on 23 February 2022. EMBASE search was limited to the English language, and studies indexed as “case report”, “review”, “meta-analysis” or “observational” were excluded. PEDro and Cochrane CENTRAL searches were limited to clinical trials only. Full search strategy for each database is presented in Appendix A. In addition, the reference lists of included studies were screened for additional studies.

### 2.2. Study Eligibility

Inclusion and exclusion criteria are presented below, according to the PICOS framework (complete list is presented in Appendix A).

*Population*: This review included studies which involved patients with advanced or metastatic cancer. For the purposes of this review, advanced cancer was defined by either the study authors describing the population as ‘advanced’, ‘locally advanced’, or stage III or above. In studies that included mixed populations, such as both non-advanced cancer and advanced cancer, the study was included if the results were appropriately stratified by disease stage or if the population was composed of at least 75% patients with advanced/metastatic cancer. 

*Intervention:* This review included interventional studies with an exercise training component, either with or without nutritional intervention such as nutrition counselling, prescribed meal plans, or supplementation. For this study, ‘exercise training’ was taken to include any intervention that was longer than a single session and aimed to increase an individual’s physical activity through aerobic exercise, resistance exercise, or sport. Interventions that provided exercise or physical activity recommendations without explicit exercise prescription were included only if the manuscript reported a measure of adherence. Nutritional interventions alone were not included. 

*Control:* This review included all studies with a control or comparator, which may itself be a different kind of intervention, provided that the patients involved were comparable to those in the intervention group, i.e., had advanced or metastatic cancer. Healthy participants as the comparator group were excluded. 

*Outcomes:* This review included all studies that reported the effect of a relevant intervention on body composition, including any constituent of lean or fat mass (e.g., lean body mass, skeletal muscle mass, muscle cross sectional area, fat mass). This review excluded studies that only reported undifferentiated body mass, body mass index, or bone mass outcomes. 

*Study type*: This review included all prospective randomized and non-randomized controlled trials, and therefore excluded retrospective and single-arm studies. 

### 2.3. Study Selection

Titles and abstracts from the database searches were imported into Covidence systematic review software (Veritas Health Innovation, Melbourne, Australia). Each abstract was screened twice by a team of independent reviewers (OB, RW, PGE, DK, CC, TB) with any conflict resolved through group discussion. The full texts were then also reviewed twice by a team of independent reviewers (OB, RW, PGE, DK, CC, TB), with any conflict resolved through group discussion. OB screened every abstract and full text. If consensus could not be agreed by group discussion, the corresponding author (CDC) was available to aid decision making. 

### 2.4. Data Extraction 

Data from each included study were extracted using a predefined template in Covidence that included study details and background, participant characteristics, flow of participants through the trial, treatment of the intervention and comparator groups, body composition outcomes measured (including any quantitative non-significant and significant between- and within-group results reported), and adverse events.

### 2.5. Methodological Assessment 

Risk of bias for randomized trials was independently assessed by two reviewers (DK, TB) within Covidence using the Cochrane risk-of-bias tool for randomized trials [65]. Risk of bias for non-randomized controlled trials was assessed using a separate spreadsheet, using the Cochrane risk-of-bias tool for non-randomized studies (ROBINS-I) [66]. 

## 3. Results

### 3.1. Study Characteristics 

A total of 6092 records were identified from seven databases, leaving 4528 records after duplicate removal. Of these, 20 studies fit the inclusion criteria and were included in the systematic review (Figure 1). 

Of the twenty studies included in this systematic review, sixteen (80%) describe original results from randomized-controlled trials (RCTs) [48,67,68,69,70,71,72,73,74,75,76,77,78,79,80,81], three (15%) from non-randomized controlled trials [82,83,84], and one (5%) was a secondary analysis [85] of another [71]. The articles included within this analysis were published between 2013 and 2022 and conducted across the globe including Denmark (*n* = 4) [67,74,78,79], Germany (*n* = 3) [75,83,84], the United Kingdom (*n* = 3) [71,81,85], Australia (*n* = 3) [48,69,73], the United States (*n* = 2) [80,82], Switzerland (*n* = 2) [68,77], Norway (*n* = 2) [71,85], Taiwan (*n* = 1) [70], India (*n* = 1) [72], and the Netherlands (*n* = 1) [76]. 

In total, 1091 patients were allocated to the interventions of the included studies with an average age ranging from 44.0 [72] to 76.9 years old [48] and where six studies (30%) exclusively examined males [48,67,69,73,74,78] and two studies (10%) examined females [72,80]. Twelve studies (60%) exclusively targeted specific cancer types or anatomical regions, which included six (30%) examining prostate cancer [48,67,69,73,74,78], one (5%) examining breast cancer [80], four (20%) including patients with gastrointestinal cancers [70,75,76,81], and one (5%) including patients with head and neck cancers [82], whilst the remaining eight (40%) included mixed populations [68,71,72,77,79,83,84,85]. The cancer stages reported within the studies include six (30%) exclusively involving patients with metastatic cancer [48,69,73,76,78,80] and six (30%) examining a mixture of locally advanced, advanced, and metastatic patients [67,68,74,77,79,81], whilst six (30%) reported the inclusion of patients with stage III and IV cancer [71,75,82,83,84,85], and the remaining two studies (10%) reported the inclusion of advanced [72] or locally advanced patients [70]. 

During the respective study intervention periods, 12 studies (60%) examined patients who were actively receiving or about to receive treatment as dictated as part of their inclusion criteria. These treatments included chemotherapy (*n* = 6; 30%) [71,75,76,81,82,85], chemoradiotherapy (*n* = 1; 5%) [70], and androgen deprivation therapy (*n* = 2; 10%) [67,74], with three (15%) studies examining patients on a range of treatments (e.g., chemotherapy, immunotherapy, targeted therapy, radiotherapy, hormone therapy) [79,83,84]. Of the remaining studies, five (25%) reported some patients to be receiving treatment (e.g., anti-androgen monotherapy, castration, chemotherapy, endocrine therapy, immunotherapy) but treatment status was not an inclusion criterion [48,73,77,78,80], two (10%) did not report any active treatment [68,69], and one (5%) reported patients to be receiving palliative care but did not specify treatment regimens [72]. The detailed study and participant characteristics are summarized in Table 1, where studies are grouped by their intervention of either exercise alone, exercise plus a nutritional component, or exercise (with or without a nutritional component) with a significant other component. Other component is defined as an additional component to the exercise or exercise and nutritional intervention and may include the consumption of medications, psychological support, goal setting etc.

### 3.2. Intervention Characteristics

Within this review, the included studies are categorized into three groups based on their intervention: exercise alone (*n* = 6; 30%) [67,69,73,74,75,78], combined exercise and nutrition (*n* = 8; 40%) [70,72,76,77,82,83,84,86], and exercise with or without nutrition but containing an additional component (*n* = 6; 30%) [48,71,79,80,81,85]. The detailed intervention characteristics are summarized in Table 1. 

Regarding the exercise component, eight studies (40%) exclusively used supervised exercise [48,67,68,70,73,78,83,84], six studies (30%) used unsupervised exercise [71,72,74,75,76,85], and the remaining six (30%) utilized both supervised and self-directed exercise [69,77,79,80,81,82]. The interventions incorporated resistance training (e.g., machine weights, resistance bands) [48,68,69,71,73,77,79,80,81,82,85], aerobic training (e.g., treadmill, cycling) [48,69,71,73,77,80,81,85], high intensity interval training [48], football training [67,78], gaming console-based exercise [74], walking [70,75,79,82], self-directed physical activity [72,76], and electromyostimulation with additional exercises [83,84]. The intervention duration ranged from 4 weeks [70] to 6 months [48,72,78], with the most common duration being 12 weeks/3 months [67,68,69,73,74,75,77,79,80,83,84]. The exercise frequency prescribed most often was two sessions per week (*n* = 7; 35%) [48,68,69,78,79,83,84]; however, another seven studies (35%) reported varying frequencies that either increased across the intervention [67,82], depended on participant choice [75], or varied by exercise mode (e.g., aerobic exercise twice a week and resistance three times a week, or two supervised, one home-based session per week) [71,77,81,85]. The remaining studies prescribed between 3 and 5 sessions per week [70,73,74,76,80], or did not prescribe a frequency [72]. The intensity of these sessions was typically reported as moderate using either a percentage of maximal heart rate [70,73,81], repetition maximum [48,68,73,77,79,82], or the Borg scale of perceived exertion [75,77,81]. 

In terms of nutritional interventions, nutrition counselling occurred either at baseline only [71,83,84,85], a minimum of three times throughout the intervention period [68,77], or at regular intervals including weekly [70,76,80] or biweekly [72]. Alongside nutritional counselling, several studies included nutritional supplements such as a flour mixture (roasted bengal gram flour, roasted barley flour, roasted soybean flour, flaxseed powder, dried amaranthus spinosus powder) [72], protein supplement [68,77,79], or an oral nutrition supplement including eicosapentaenoic acid [71,85]. Additionally, other studies prescribed a specific nutrition goal of >1.0 g/kg of protein per day and a minimum intake of 25 kcal/kg/d [83,84], >1.2 g/kg of protein per day [76], or whole grains and five or more fruit and vegetables per day [80]. Furthermore, many studies including a nutrition component also reported some degree of nutritional counselling in the usual care control group as part of standard of care protocols [72,76,77,82,83,84], although this was typically less formalized or intensive compared to intervention. 

Regarding the further components that studies included alongside exercise and nutritional interventions, several utilized psychological counselling/support [48,79,81], whilst others included daily intake of the anti-inflammatory celecoxib [71,85], or an intervention rooted in social cognitive theory including lifestyle counselling [80].

### 3.3. Adherence and Adverse Events

Adherence to supervised exercise sessions ranged from 54.0% [78] to 93.0% [69,80] and self-directed exercise ranged from 16.0% [76] to 95.0% [77]. Adherence to components of the nutrition intervention ranged from 48.0% [71] to 100% [68] for nutritional supplements, 84.0% [80] to 106.7% [77] for nutrition counselling, and 49.0% [76] to 74.2% [83] for achieving nutrition-related goals. Of the fifteen studies that recorded adverse events, eight reported no adverse events [68,69,73,75,80,81,82,84] and seven reported some adverse events [67,71,74,77,78,79,83]. These were mostly minor effects such as muscle soreness [83], or events that also occurred in the usual care group and so were unlikely to be related to the intervention, such as falls [78] or nausea and pain [71,74,77]. Two studies reported fractures that may have been related to the intervention but were not at sites of known bone metastases [67,79]. 

### 3.4. Outcome Measures 

Body composition was identified as the primary outcome in six studies (30%) [67,70,72,76,83,85]. Body composition was measured by bioelectrical impedance analysis (BIA) (N = 8; 40%) [68,70,74,75,77,79,83,84], dual energy X-ray absorptiometry (DXA) (N = 8; 40%) [48,67,69,73,78,79,80,82], computed tomography (N = 4; 20%) [71,76,81,85], and caliper assessment of skinfolds (N = 1; 5%) [72]. Importantly, one study utilized both BIA and DXA yet reported significant results by DXA but not by BIA [79]. Additionally, eight studies (40%) explicitly stated that body composition measures were conducted by a blinded assessor [71,76,78,79,80,81,82,85]. The detailed outcome measures are summarized in Table 1. 

#### 3.4.1. Lean Mass-Related Outcomes

Six exercise only studies reported a lean mass-related outcome [67,69,73,74,75,78] with three reporting a significant effect [67,69,75]. One study [69] reported a significant between-group difference for both lean mass (kg) (1.7 kg) and appendicular lean mass (kg) (1.0 kg), where the exercise group exhibited a small increase in both outcomes compared to a loss in the control group. A separate study [75] reported a significant between-group difference, with a greater percent increase for lean mass in the exercise group compared to control (3.40 vs 0.64%). Finally, a third study [67] reported a significant within-group increase in lean mass in the exercise group of 0.5 kg, which was significantly different to that in the control group by 0.7 kg. 

Five studies examined the effect of a combined exercise and nutritional intervention on lean mass-related outcomes [70,77,82,83,84], of these, two similar non-randomized studies by the same research group [83,84] reported significant between-group differences in skeletal muscle mass of 0.53 kg [83] and 0.99 kg [84]. Finally, among the five studies incorporating an additional component and measuring lean mass-related outcomes [48,71,79,80,85], three reported a significant effect [79,81,85]. One study reported a significant between-group difference where the exercise group had less skeletal muscle loss (cm^2^/m^2^) than the control group (−11.6 versus −15.6 cm^2^/m^2^) [81]. A separate study [79] reported a significant within-group increase in lean mass (kg) in the exercise group (47.3 to 48.7 kg) and a significant between-group difference when compared to the control group (0.9 kg). Finally, another study [85] reported a significant within-group decline in skeletal muscle index (cm^2^/m^2^) in the control group (−1.8 cm^2^/m^2^). 

#### 3.4.2. Fat Mass-Related Outcomes

Among the five exercise-only studies reporting fat mass-related outcomes, no significant effects were reported [67,69,73,74,78]. Three combined exercise and nutrition studies reported fat mass-related outcomes [72,77,83]: a significant between- and within-group change in body fat percent was reported in one study, where the intervention group increased body fat percent from 20.5 to 23.7% and the control group declined from 25.4 to 24.5% [72]. Finally, among the four exercise studies with an additional component reporting a fat mass-related outcome [48,79,80,85], one study [80] reported a significant within-group decline in visceral fat mass for both the intervention (−99 g) and control (−81 g) groups, and a significant between-group difference (−89 g) signifying more fat loss in the intervention group. 

### 3.5. Risk of Bias Assessment 

The results of the risk of bias assessment for each included study are summarized in Figure 2. In brief, for 17 RCTs included in this review, the risk of (a) sequence generation was low in 16 studies (94%) and unclear in 1 study (6%); (b) allocation concealment was low in 9 studies (53%) and unclear in 8 studies (47%); (c) blinding of participants and personnel was high in all studies (100%); (d) blinding of outcome assessment was low in 13 studies (76%), unclear in 3 studies (18%), and high in 1 study (6%); (e) incomplete outcome data was low in 16 studies (94%) and high in 1 study (6%); (f) selective reporting was low in 13 studies (76%), high in 3 studies (18%), and unclear in 1 study (6%); and (g) other sources of bias were low in 12 studies (71%), unclear in 4 studies (23%), and high in 1 study (6%). For the three non-RCTs, the overall risk of bias was serious in two studies (67%) and moderate in one study (33%). 

## 4. Discussion

To the authors’ knowledge, this is the first systematic review to provide a comprehensive summary of the effects of exercise with or without nutritional interventions on lean mass and fat mass among patients with advanced or metastatic cancer. Twenty studies were considered eligible, and despite the inclusion of a heterogeneous sample, there was a trend for both exercise alone and combined exercise and nutritional interventions to preserve or improve lean mass-related outcomes. Moreover, only those studies utilizing combined exercise and nutritional interventions reported alterations to fat mass-related outcomes.

Our systematic review indicates that both exercise and combined exercise and nutrition interventions have some impact on lean mass-related outcomes among patients with advanced or metastatic cancer, where 50% of studies measuring a lean mass-related outcome reported a significant intervention effect [67,69,75,79,81,83,84,85]. However, the fact that 50% of the studies reported no significant intervention effect means the overall effect is unclear, although this may relate to the low statistical power of many studies. Nonetheless, every study reporting a significant effect reported relative maintenance or gain of lean mass in the intervention group compared to control. This trend favoring the intervention groups suggests a possible effect of exercise both with and without a nutritional component and builds upon congruent findings in existing reviews [44,57,58]. The combination of resistance training and increased protein intake has been associated with lean mass preservation or increased muscle protein synthesis among patients with cancer [88,89,90]. In our included population, only one study incorporating resistance training, alongside protein intake [79], found a significant mean increase in lean mass in the intervention group compared to a 0.4 kg non-significant maintenance in the control group across a 12-week period. Schink et al. [83,84], who noted a significant improvement in lean mass-related outcomes, also encouraged an increase in protein intake; however, they utilized an electromyostimulation exercise protocol in both studies. The final two studies which combined protein supplementation with resistance training found no effect on lean mass-related outcomes [68,77]; notably, neither of these studies were powered to detect changes in body composition. Football training and walking were also demonstrated to improve lean mass [67,75] with Uth et al. [67] reporting a significant within-group increase in lean mass (0.5 kg) in patients with prostate cancer compared to a non-significant maintenance in the control group (−0.2 kg), whilst Stuecher et al. [75] reported retention of lean mass % with a walking intervention in patients with gastrointestinal cancer. A limitation of our systematic review is the breadth of inclusion criteria and heterogeneous nature of intervention protocols, as such, no conclusion can be proposed as to the best exercise or nutrition prescription to achieve lean mass preservation or development. Nonetheless, the encouragement of movement, even if it is simply walking [75], and the optimization of protein intake are likely beneficial [49] given their demonstrated potential and tolerability.

Only two (17%) studies that reported a fat mass-related outcome noted a significant intervention effect [72,80]. Furthermore, these two studies incorporated both an exercise and nutrition component. Kapoor et al. [72] reported that the intervention group showed an increase in fat mass compared to controls, while Sheean et al. [80] reported a decrease in visceral adipose tissue mass in the intervention group compared to the control. In the context of advanced and metastatic disease, these fat mass-related results are not necessarily conflicting, and instead may reflect differing aims, populations, and intervention characteristics between the two studies. Kapoor et al. [72] focused on a population with cachexia receiving palliative care and encouraged the consumption of energy dense food with a minor physical activity component, where they considered a gain in fat mass to be desirable. In contrast, Sheean et al. [80] focused on patients with clinically stable metastatic breast cancer with no pre-existing unintentional weight loss who had a baseline mean body mass index of 29.35 kg/m^2^ and encouraged a high volume of exercise (4 days/week, 150 min/week) alongside the consumption of whole grains, fruits, and vegetables. Notably, Sheean et al. [80] also measured total body fat mass (kg) and body fat percent (%), showing no significant within- or between-group differences. However, this loss of visceral fat, the more metabolically active fat depot, may have implications on metabolic health such as improving insulin resistance and chronic inflammation [91,92]. Our fat mass-related findings build on those of previous reviews, which similarly conclude there to be uncertainty about the impact of exercise interventions in substantially altering fat mass among patients with advanced cancer [57]. Nevertheless, the direction of fat mass changes depends on context; in advanced and metastatic patients, particularly those who are cachexic, fat loss may not be a desirable outcome as outlined by Kapoor et al. [72]. Therefore, like lean mass-related outcomes, a conclusion regarding the recommended exercise and nutrition prescription for advanced and metastatic patients cannot be provided as it should be goal-specific (fat loss, gain, or maintenance) and population-specific. However, if fat mass is the desired body composition outcome to be targeted in a specific advanced/metastatic cancer population, the inclusion of a combined exercise and nutrition intervention, as opposed to exercise alone, will likely have a greater impact [36]. 

Future studies should continue to quantitatively assess the extent to which exercise, with or without a nutritional intervention or ‘other’ intervention components as detailed in Table 1, affects both lean and fat mass. Although the “other” components were not discussed in the current review in the context of their influence on body composition due to the variation of strategies (e.g., medications, psychological counselling, goal setting, etc.), it is possible that these additional intervention components may help to induce more behavioral changes (e.g., through education/counselling) or physiological changes (e.g., prescribing medications), than exercise or nutrition alone. As such, these components should also be considered in future studies in the context of body composition manipulation. Furthermore, it is also important to note that bone mass was not included as an outcome in this review because of the potential influence of bone metastases on bone mass which would be difficult to differentiate and account for in a systematic review analysis. However, this is an interesting topic for future reviews due to the interaction between bone metabolism, androgen-deprivation, and treatment-induced menopause and should be pursued when a larger volume of studies have examined the effects of exercise and nutrition on bone mass among those with bone metastases [93]. While caution is warranted in the conclusions provided by this systematic review due to the heterogeneous nature of the included studies, this collation of exercise interventions with or without a nutritional component may also help inform clinical practice. Alongside somewhat promising effects on body composition, the included studies generally had few adverse effects and were well-tolerated. Hence, the evidence presented here supports current clinical practice guidelines from ESMO [51], ASCO [54], and ESPEN [52,53], and may contribute to their use in clinical settings. 

A strength of this review is that it is the first to comprehensively examine exercise and combined exercise and nutritional interventions in patients with advanced and metastatic cancer beyond a single cancer type. Until now, nutrition as an additive component to exercise had not been addressed in a systematic review among patients with advanced or metastatic cancer. We acknowledge the additional benefit nutrition may have in an exercise intervention, with both components possibly acting in synergy and now recognized as critical in clinical practice [51,52,53,54]. However, this review has several limitations. First, the included studies are diverse in their included populations, intervention characteristics, and outcomes measured. This means it is not possible to draw specific conclusions about the true quantitative effects of the included interventions, such as through meta-analysis, nor is it feasible to directly compare many of the included studies with each other. In addition, the results of some of the included studies must be interpreted with caution. For example, several studies reported related or overlapping body composition outcomes (e.g., measured using both DXA and BIA) [71,79,85] increasing the likelihood of positive results due to random chance as noted by Mikkelsen et al. [79]. Eight of our included studies measured body composition by BIA [68,70,74,75,77,79,83,84]. Importantly, BIA is an indirect way to measure body composition, dependent on assumptions derived from population means [94] that can also be strongly influenced by hydration status, which can vary significantly in patients with cancer [95]. One included study even showed that patients in the intervention group had significantly lower hydration status than controls after 12 weeks, which may have affected results [83]. Likewise, BIA can be subject to large errors when predicting fat free mass in patients with low % body fat [94]. Importantly, many of the included studies were too underpowered to detect significant body composition results [67,68,69,70,71,73,74,77,82], whilst others based their power calculations on other outcomes [48,72,74,75,79,81] or reported no power calculations [78,80,83,84]. This largely reflects a high number of pilot studies in a field which faces difficulty in recruitment and retention. 

Another limitation, as noted by other reviews [55,58], is the inclusion of studies that had a stringent criteria for inclusion, such as current treatment status, which selects patients that are well enough and motivated enough to participate in the intervention, possibly reducing applicability to the overall population of patients with advanced or metastatic cancer. Locally advanced cancers were also included in this review to provide a comprehensive summary, but this warrants caution when interpreting the intervention effect as a patient with locally advanced cancer may be fitter or healthier and therefore more able to complete a higher intensity or volume of exercise and induce greater changes leading to skewed data in a mixed population. Cancers affecting different body sites may vary significantly in the extent to which they impact body composition. For example, even locally advanced gastrointestinal cancers may affect body composition more than other cancer types that have metastasized, principally through the direct effects of the former on nutritional absorption. Further caution is recommended when interpreting the results in the long term, given that the follow-up periods were generally short and adherence rates were low in some studies. Furthermore, many studies included some degree of nutritional intervention as part of standard care in the control group [72,76,77,82,83,84], although this was typically less intensive than the nutritional intervention, this should be taken into context when considering the true effect of interventions.

## 5. Conclusions

This review provides a summary of the effects of exercise with and without nutritional interventions on body composition in patients with advanced and metastatic cancer. Our review indicates that exercise interventions with and without nutrition may preserve or improve lean mass-related outcomes among patients with advanced and metastatic cancer. Moreover, only combined exercise and nutrition interventions induced alterations in fat mass-related outcomes, although further studies are clearly needed. Whilst unable to definitively outline the extent to which exercise with or without nutritional interventions can alter lean and fat mass, our findings summarize the ongoing research in this field and suggest that future studies should investigate clear hypotheses about their intended effects on body composition. A better understanding of the role of body composition in advanced and metastatic cancer will benefit our understanding of cancer survivorship as a whole, particularly with respect to improved quality of life and tolerance of ongoing treatment. In turn, as exercise and nutrition-related research continues to develop in this understudied population, the inclusion of such strategies should be considered more frequently in oncologic care in line with emerging guidelines.

## Figures and Tables

**Figure 1 nutrients-14-02110-f001:**
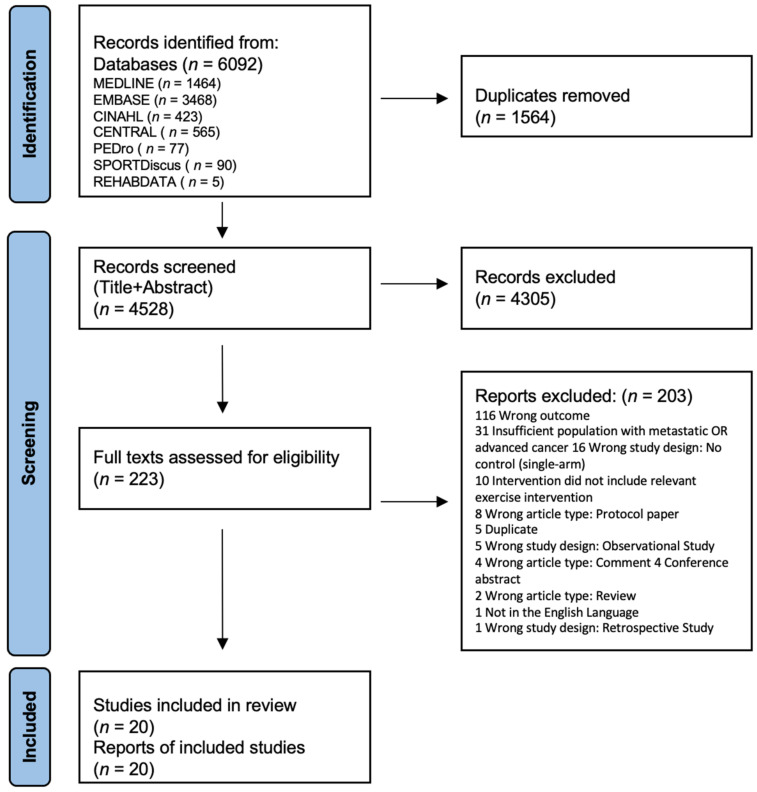
Flowchart detailing study selection.

**Figure 2 nutrients-14-02110-f002:**
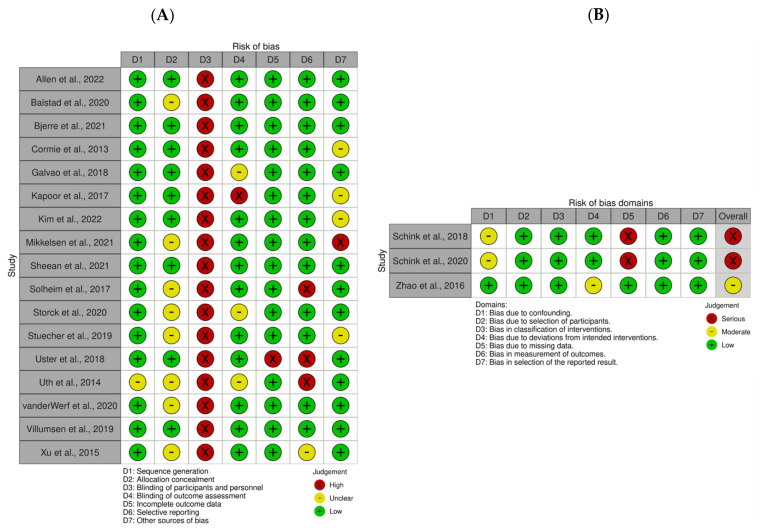
Risk of bias assessments for (**A**) randomized controlled trials and (**B**) non-randomized controlled trials. Risk of bias plots were generated using the Risk-of-bias VISualization (robvis) tool [48,67,68,69,70,71,72,73,74,75,76,77,78,79,80,81,82,83,84,85,87].

**Table 1 nutrients-14-02110-t001:** Summary characteristics and results of each study.

Study Details	Population	Experimental Groups	Intervention	Body Composition Outcomes	Pre vs. Post Training Mean ± SD or MD ± SD/(95%CI)
* **Exercise only** *					
Cormie et al. [69] **Australia** RCT	**Population:** Prostate cancer with bone metastases (*n* = 20) **Stage**: Metastatic: 100%	**EX** (*n* allocated = 10; *n* completed outcome = 8): Supervised resistance training, home-based aerobic training. **CON** (*n* allocated = 10; *n* completed outcome = 7): Self-directed exercise	12-week intervention **Exercise component (FITT):** *F*: RT = 2 days/week; AT = NR. **I**: RT = 8–12 RM; AT = moderate intensity. **Time**: RT = 60 min, 8–12 reps, 2–4 sets; AT = 150 min/week. **Type**: Supervised machine weight resistance training that did not target areas of bone metastases, self-directed home-based aerobic exercise. **Adherence**: attended 93% of supervised exercise sessions.	Lean mass (kg) (DXA)	Within-group differences: **EX**: PRE: 57.2 ± 7.8 vs. POST: 57.8 ± 8.0 **CON**: PRE: 53.2 ± 9.7 vs. POST: 52.5 ± 8.0 Between-group differences: **EX vs. CON**: MD: 1.7 (0.2 to 3.2) ¥↑
				Appendicular lean mass (kg) (DXA)	Within-group differences: **EX**: PRE: 24.3 ± 3.7 vs. POST: 24.5 ± 3.7 **CON**: PRE: 21.4 ± 3.9 vs. POST: 20.9 ± 3.3 Between-group differences: **EX vs. CON**: MD: 1.0 (0.4 to 1.6) ¥↑
				Fat mass (kg) (DXA)	Within-group differences: **EX**: PRE: 27.7 ± 5.6 vs. POST: 27.8 ± 6.0 **CON**: PRE: 27.2 ± 5.7 vs. POST: 27.5 ± 6.5 Between-group differences: **EX vs. CON**: MD: −0.3 (−1.4 to 0.9)
				Trunk fat mass (kg) (DXA)	Within-group differences: **EX**: PRE: 14.7 ± 3.4 vs. POST: 14.6 ± 3.7 **CON**: PRE: 15.0 ± 3.4 vs. POST: 15.0 ± 3.8 Between-group differences: **EX vs. CON**: MD: 0.0 (−0.6 to 0.6)
				Visceral fat mass (kg) (DXA)	Within-group differences: **EX**: PRE: 0.89 ± 0.20 vs. POST: 0.89 ± 0.23 **CON**: PRE: 0.96 ± 0.19 vs. POST: 0.96 ± 0.19 Between-group differences: **EX vs. CON**: MD: 0.01 (−55.3 to 58.6)
				Body fat percent (%) (DXA)	Within-group differences: **EX**: PRE: 31.7 ± 4.9 vs. POST: 31.5 ± 5.1 **CON**: PRE: 32.7 ± 2.2 vs. POST: 33.0 ± 3.3 Between-group differences: **EX vs. CON**: MD: −0.4 (−1.9 to 1.2)
Uth et al. [67] **Denmark** RCT	**Population**: Advanced or locally advanced prostate cancer (*n* = 57) **Stage**: ≥T3: 70.2%	**EX** (*n* allocated = 29, *n* completed outcome = 26): Supervised football training **CON** (*n* allocated = 28, *n* completed outcome = 23): Usual care control.	12-week intervention **Exercise component (FITT):** **F**: 2–3 days/week **I**: Not prescribed but a mean HR of 84.6 ± 3.9% of individual max HR was achieved. **Time**: 45 min **Type**: Football drills and game. **Adherence**: attended 76.5 ± 24.2% of supervised exercise sessions.	Lean mass (kg) (DXA)	Within-group differences: **EX**: MD: 0.5 (0.1 to 0.9) ¥↑ **CON**: MD: −0.2 (−0.6 to 0.2) Between-group differences: **EX vs. CON**: MD: 0.7 (0.1 to 1.2) ¥↑
				Fat mass (kg (DXA)	Within-group differences: **EX**: MD: −0.6 (−1.4 to 0.1) **CON**: MD: 0.0 (−0.5 to 0.5) Between-group differences: **EX vs. CON**: MD: −0.6 (−1.5 to 0.2)
				Body fat percent (%) (DXA)	Within-group differences: **EX**: MD: −0.7 (−1.3 to 0.0) **CON**: MD: 0.1 (−0.4 to 0.5) Between-group differences: **EX vs. CON**: MD: −0.7 (−1.5 to 0.2)
Galvao et al. [73] **Australia** RCT	**Population:** Prostate cancer with bone metastases(*n* = 57) **Stage**: Metastatic: 100%	**EX** (*n* allocated = 28, *n* completed outcome = 23): Supervised aerobic and resistance exercise. **CON** (*n* allocated = 29, *n* completed outcome = 26): Usual care control.	12-week intervention **Exercise component (FITT):** **F**: 3 days/week **I**: AT = 60–85% HRmax; RT = 10–12 RM **Time**: 60 min sessions; AT = 20–30 min; RT = 10–12 reps, 3 sets. **Type**: Exercises did not target bone metastases sites. AT = choice or walking, cycling, rowing; RT = machine based. **Adherence**: attended 89% of supervised exercise sessions.	Lean mass (kg) (DXA)	Within-group differences: **EX**: PRE: 56.6 ± 8.1 vs. POST: 56.2 ± 8.0 **CON**: PRE: 55.6 ± 7.8 vs. POST: 55.4 ± 7.5 Between-group differences: **EX vs. CON**: MD: −0.3 (−1.3 to 0.7)
				Fat mass (kg) (DXA)	Within-group differences: **EX**: PRE: 28.7 ± 8.1 vs. POST: 29.0 ± 7.8 **CON**: PRE: 28.3 ± 6.9 vs. POST: 29.0 ± 6.4 Between-group differences: **EX vs. CON**: MD: −0.2 (−1.2 to 0.7)
Villumsen et al. [74] **Denmark** RCT	**Population:** Locally advanced or advanced stage prostate cancer (*n* = 46) **Stage**: Bone metastases: 34.8% Lymph node metastases: 6.5%	**EX** (*n* allocated = 23, *n* completed outcome = 21): Home-based exergaming **CON** (*n* allocated = 23, *n* completed outcome = 20): Usual care control inclusive of physical activity advice.	12-week intervention **Exercise component (FITT):** **F**: 3 days/week **I**: NR **Time**: 60 min **Type**: Exergaming using both aerobic and strength exercises, free weights. **Adherence**: Completed on average 153.5 min/week from a prescribed 180 min/week.	Lean mass (%) (BIA)	Within-group differences: **EX**: NR **CON**: NR Between-group differences: **EX vs. CON**: MD: 0.91 (−0.2 to 2.0)
				Fat mass (% (BIA)	Within-group differences: **EX**: NR **CON**: NR Between-group differences: **EX vs. CON**: MD: −0.9 (−2.0 to 0.2)
Stuecher et al. [75] **Germany** RCT	**Population:** Stage III or IV gastrointestinal tract cancers(*n* = 44) **Stage**: Metastatic: NR	**EX** (*n* allocated = 22, *n* completed outcome = 13): Self-directed walking. **CON** (*n* allocated = 22, *n* completed outcome = 15): Usual care control.	12-week intervention **Exercise component (FITT):** **F**: 3–5 days/week **I**: 11–13 RPE **Time**: 150 min/week **Type**: Home-based walking. **Adherence**: 81.3% completed the home-based program as prescribed.	Lean mass (%) (BIA)	Within-group differences: **EX**: MD: 3.4 ± 4.6 **CON**: MD: 0.64 ± 3.4 Between-group differences: **EX vs. CON**: MD: NR, *p* = 0.02. ¥↑
				Phase angle (°) (BIA)	Within-group differences: **EX**: MD: 0.13 ± 0.91 **CON**: MD: −0.01 ± 0.69 Between-group differences: **EX vs. CON**: MD: NR, *p* = 0.2
Bjerre et al. [78] **Denmark** RCT	**Population:** Prostate cancer with bone metastases (*n* = 41) **Stage**: Metastatic: 100%	**EX** (*n* allocated = 22, *n* completed outcome = 21): Community-based football intervention **CON** (*n* allocated = 19, *n* completed outcome = 15): Usual care	6-month intervention **Exercise component (FITT):** **F**: 2 days/week **I**: NR **Time**: 60 min **Type**: Supervised group-based football training involving bodyweight training, football skills and football match play. **Adherence**: attended 63% of supervised group sessions (at week-12); attended 54% of supervised group sessions (at 6-months).	Lean mass (kg) (DXA)	Within-group differences: **EX**: MD: −0.3 (−1.1 to 0.5) **CON**: MD: −0.4 (−1.3 to 0.6) Between-group differences: **EX vs. CON**: MD: −0.2 (−1.4 to 0.9)
				Fat mass (kg) (DXA)	Within-group differences: **EX**: MD: −0.4 (−1.3 to 0.6) **CON**: MD: −0.2 (−1.4 to 1.0) Between-group differences: **EX vs. CON**: MD: 0.4 (−1.1 to 1.8)
* **Combined exercise and nutrition** *					
Xu et al. [70] **Taiwan** RCT	**Population:** Locally advanced tumors of the esophagus (*n* = 56) **Stage**: Stage 1: 3.6% Stage 2: 7.1% Stage 3: 82.1%	**EX + NU** (*n* allocated = 28, *n* completed outcome = 28): Supervised walking and nutrition counselling. **CON** (*n* allocated = 28, *n* completed outcome = 28): Usual care control.	4–5-week intervention **Exercise component (FITT):** **F**: 3 days/week **I**: 60% age predicted maximum HR **Time**: 25 min **Type**: Walking **Nutrition component:** Weekly nutrition counselling. **Adherence:** EX: Completed 8.4 ± 3.6 of supervised walking sessions.NU: attended 100% of nutrition sessions.	Lean mass (kg) (BIA)	Within-group differences: **EX + NU**: MD: −0.7 ± 1.9 **CON**: MD: −2.0 ± 3.0 Between-group differences: **EX + NU vs. CON**: MD: 1.3 (−0.05 to 2.66)
Kapoor et al. [72] **India** RCT	**Population:** Females with advanced cancer (*n* = 63) **Stage**: NR	**EX + NU** (*n* allocated = 30, *n* completed outcome = 17): Multimodal (Nutrition counselling, oral nutrition supplement, physical activity recommendation) **CON** (*n* allocated = 33, *n* completed outcome = 15): Nutrition counselling and physical activity recommendation	6-month intervention **Exercise component (FITT):** **F**: NR **I**: Not prescribed but reported: **EX + NU**: PRE: 33.6 ± 3.9 METs vs. POST 31.9 ± 2.7 METs (*p* = 0.274); **CON**: PRE: 30.7 ± 2.7 METs vs. POST 28.0 ± 2.5 METs (*p* = 0.004). **Time**: NR. **Type**: Low levels of PA, e.g., walking and participation in household activities. **Nutrition component:** Bi-weekly nutrition counselling visits. 100 g/day of IAtta oral nutrition supplement (mixture of roasted bengal gram flour, roasted barley flour, roasted soybean flour, flaxseed powered, dried amaranthus spinosus powder). **Adherence**: EX: NR. NU: NR. EX + NU: 51% completed the intervention as prescribed.	Body fat percent (%) (skinfolds)	Within-group differences: **EX + NU**: PRE: 20.5 ± 5.2 vs. POST: 23.7 ± NR ¥↑ **CON**: PRE: 25.4 ± 6.5 vs. POST: 24.5 ± NR ¥↓ Between-group differences: **EX + NU vs. CON**: MD: NR; *p* = 0.001 ¥↑
Uster et al. [68] **Switzerland** RCT	**Population:** Metastatic or locally advanced tumors of gastrointestinal and lung tracts (*n* = 58) **Stage**: Stage III: 2% Stage IV: 98%	**EX + NU** (*n* allocated = 29, *n* completed outcome = 24): Multimodal (Supervised group-based resistance and balance training, nutrition counseling) **CON** (*n* allocated = 29, *n* completed outcome = 20): Usual care control.	3-month intervention **Exercise component (FITT):** **F**: 2 days/week **I**: RT= 60–80% of 1-RM; Balance = NR. **Time**: 60 min, RT = 10 reps, 2 sets, Balance= 1–2 min per move. **Type**: RT = resistance machines; balance mat. **Nutrition component:** Minimum of 3 nutritional counselling during intervention encouraging patients to consume 1.2 g protein/kg body weight/day, with emphasis on consuming protein after exercise sessions. **Adherence**: **EX**: attended 67% of supervised exercise sessions. NU: 89.7% completed the minimum nutritional counseling sessions. EX + NU: 100% consumed at least 9–10 g of protein after each exercise session.	Phase angle (°) (BIA)	Within-group differences: E**X + NU**: NR. **CON**: NR. Between-group differences: **EX + NU vs. CON**: MD: NR.
Zhao et al. [82] **United States of America** Non-RCT	**Population:** Stage III-IV Head and neck squamous cell carcinoma (*n* = 20) **Stage**: Stage III: 22% Stage IV: 78%	**EX + NU** (*n* = 11): Multimodal (Supervised and unsupervised aerobic and resistance training, nutrition counselling) **CON** (*n* = 7): Standard of care inclusive of nutritional counselling.	14-week intervention (7 weeks supervised, 7 weeks unsupervised) **Exercise component (FITT):** **F**: Supervised period = 3 days/week; unsupervised period = 5 days/week **I**: 11–13 RPE **Time**: 60 min sessions; AT = 30 min; RT = 8–12 reps, 3 sets. **Type**: AT = walking; RT = free weights **Nutrition component:** Baseline nutrition counselling. **Adherence**: EX: attended 72% of supervised exercise sessions. NU: NR.	Lean mass (%) (DXA)	Within-group differences: **EX + NU**: MD: 7 weeks: 0.2 ± 0.5 vs. 14 weeks: 4.7 ± 1.5 **CON**: MD: 7 weeks: 1.0 ± 0.7 vs. 14 weeks: 4.0 ± 0.9 Between-group differences: **EX + NU vs. CON**: NR; *p* > 0.05.
Schink et al. [83] **Germany** Non-RCT	**Population:** Advanced solid tumours (*n* = 131) **Stage**: Stage III: 26% Stage IV: 74%	**EX + NU** (*n* allocated = 96; *n* completed outcome = 58): Multimodal (supervised whole-body electromyostimulation, nutrition counselling) **CON** (*n* allocated = 35; *n* completed outcome = 27): Usual care control with nutrition counselling.	12-week intervention **Exercise component (FITT):** **F**: 2 days/week **I**: 85 Hz, 350 μs inducing a 6 s stimulation and 4 s rest. **Time**: 12–20 min, 6 reps per min. **Type**: whole-body electromyostimulation with additional light exercises. **Nutrition component:** Nutrition counselling encouraging >1 g/kg day of protein and minimum energy intake of 25 kcal/kg/day. **Adherence**: EX: attended 86.6 ± 10.8% of supervised sessions. NU: EX + NU = 67.4% and CON = 69% consumed the protein intake recommendation or more. EX + NU =74.2% and 75.8 consumed the kcal intake recommendations.	Skeletal muscle mass (kg) (BIA)	Within-group differences: **EX + NU: NRCON**: NRBetween-group differences: **EX + NU vs.****CON**: MD: 0.53 (0.05 to 0.98) ¥ ↑
				Fat mass (%) (BIA)	Within-group differences: **EX + NU**: NR **CON**: NRBetween-group differences: **EX + NU vs. CON**: MD: 0.51 (−0.46 to 1.47)
				Phase angle (°) (BIA)	Within-group differences: **EX + NU: NR** **CON**: NR Between-group differences: **EX + NU vs. CON**: MD: 0.07 (−0.06 to 0.19)
Schink et al. [84] **Germany** Non-RCT	**Population:** Advanced solid tumours (*n* = 80) **Stage**: Stage III: 24.4% Stage IV: 75.6%	**EX + NU** (*n* allocated = 58; *n* completed outcome = 26): Multimodal (supervised whole-body electromyostimulation, nutrition counselling) **CON** (*n* allocated = 22; *n* completed outcome = 15): Usual care control with nutrition counselling.	12-week intervention **Exercise component (FITT):** **F**: 2 days/week **I**: 85 Hz, 350 μs inducing a 6 s stimulation and 4 s rest. **Time**: 12–20 min, 6 reps per min. **Type**: whole-body electromyostimulation with additional light exercises. **Nutrition component:** Nutrition counselling encouraging >1 g/kg day and minimum energy intake of 25 kcal/kg/day. **Adherence**: EX: attended 88.9 ± 8.7% of supervised sessions. NU: NR	Skeletal muscle mass (kg) (BIA)	Within-group differences: **EX + NU**: NR **CON**: NR Between-group differences: **EX + NU vs. CON**: MD: 0.99 (0.09 to 1.90) ¥↑
van der Werf et al. [76] **Netherlands** RCT	**Population:** Metastatic colon cancer (*n* = 107) **Stage**: Metastatic: 100%	**NU + PA**: (*n* allocated = 52; *n* completed outcome T1 = 50; N completed outcome T2 = 39): Nutrition counselling and PA **CON** (*n* allocated = 55; *n* completed outcome T1 = 52; *n* completed outcome T2 = 33): Usual care inclusive of regular care dietician referral.	T0-T1 = mean 9 ± 3 weeks; T0-T2 = mean 19 ± 3 weeks **Exercise component (FITT):** **F**: 5 days/week **I**: moderate intensity **Time**: ≥30 min **Type**: self-directed PA. **Nutrition component:** Nutrition counselling with the goal of 1.2 g protein/kg body weight/day and at least ≥25 g protein per meal. **Adherence**: PA: T1 = 24%; T2 = 16% achieved PA recommendations. NU: T1 = 61%; T2 = 40% achieved protein intake recommendations. T1 = 61%; T2 = 49% achieved energy intake recommendations.	Skeletal muscle area (cm^2^) (CT)	Within-group differences: **NU + PA**: NR **CON**: NR Between-group differences: **NU + PA vs. CON**: MD: T0-T1: 0.3 (−3.5 to 4.0) vs. T1-T2: 0.3 (−3.4 to 4.0)
				Muscle density (Hounsfield units) (CT)	Within-group differences: **NU + PA**: NR **CON**: NR Between-group differences: **NU + PA vs. CON**: MD: T0-T1: 0.2 (−1.8 to 2.2) vs. T1-T2: −0.1 (−2.2 to 2.0)
Storck et al. [77] **Switzerland** RCT	**Population:** Metastatic or locally advanced cancers of the lungs, gastrointestinal tract, breast, ovarian, prostate, renal cell, bladder (*n* = 52) **Stage**: Metastatic: NR	**EX + NU** (*n* allocated = 27; *n* completed outcome = 23):Multimodal (supervised and self-directed aerobic and resistance exercise, nutrition counselling). **CON** (*n* allocated = 25; *n* completed outcome =18): Usual care inclusive of regular care nutrition counselling and physiotherapy.	12-week intervention **Exercise component (FITT):** **F**: 2 days/week supervised, 1 day/week home-based. **I**: AT = 3–5 RPE (10 borg); RT = NR. **Time**: 60–90 min; AT = NR; RT = 10–15 reps, 3 sets. **Type**: AT= bike or treadmill; RT = circuit, resistance bands. Nutrition component: Nutrition counselling at baseline, 6 weeks, 12 weeks, and as required between times. 15–30 g/day of whey protein. **Adherence**: EX: attended 70.7% of supervised sessions and completed 95% of home sessions. NU: attended 106.7% nutrition counselling sessions. 71.2% consumed the protein supplements.	Phase angle (°) (BIA)	Within-group differences: **EX + NU**: MD: 0.08 ± NR **CON**: MD: −0.04 ± NR Between-group differences: **EX + NU vs. CON**: MD: NR (−0.39 to 0.16)
				Lean mass (kg) (BIA)	Within-group differences: **EX + NU**: MD: 0.89 ± NR**CON**: MD: 0.46 ± NR Between-group differences: **EX vs. CON**: MD:NR (−2.04 to 1.18)
				Body cell mass (kg) (BIA)	Within-group differences: **EX + NU**: MD: 0.62 ± NR **CON**: MD: 0.33 ± NR Between-group differences: **EX + NU vs. CON**: MD:NR (−1.45 to 0.87)
				Fat mass (kg) (BIA)	Within-group differences: **EX + NU**: MD: 0.17 ± NR **CON**: MD: −0.38 ± NR Between-group differences: **EX + NU vs. CON**: MD:NR (−2.08 to 0.97)
* **Exercise with or without nutrition, plus an additional component** *					
Solheim et al. [71] **United Kingdom and Norway** RCT	**Population:** Stage III/IV NSCLC or inoperable pancreatic cancer (*n* = 46) **Stage**: Pancreas stage III = 20% Pancreas stage IV = 25% NSCLC stage III = 10% NSCLC stage IV = 47.5%	**EX + NU + O** (*n* allocated = 25; *n* completed outcome = 23): Multimodal (self-directed exercise, nutrition counselling, oral nutrition supplement, anti-inflammatory drug). **CON** (*n* allocated = 21; *n* completed outcome = 18): Standard of care	6-week intervention **Exercise component (FITT):** **F**: AT = 2 days/week; RT = 3 days/week. **I**: NR **Time**: AT = 30 min; RT = 20 min **Type**: AT = patient choice; AT = body weight and free weights. **Nutrition component:** Baseline nutrition counselling session. 220 mL of an oral nutrition supplement equating to 2 g/day of eicosapentaenoic acid. **Other component:** 300 mg/day of Celecoxib, an anti-inflammatory. **Adherence**: EX: attended 60% of exercise sessions. NU: 48% consumed the supplement O: 76% took the prescribed celecoxib.	Lean mass (cm^2^) (CT)	Within-group differences: **EX + NU + O: MD**: −2.82 ± 9.41 **CON**: MD: −4.97 ± 7.80 Between-group differences: **EX + NU + O vs. CON**: MD: NR
Balstad et al. [85] **United Kingdom and Norway** Secondary analysis of Solheim et al., 2017.	See Solheim et al. [71]	**EX + NU + O** (*n* allocated = 23; *n* completed outcome = 22):Multimodal (self-directed exercise, nutrition counselling, oral nutrition supplement, anti-inflammatory drug). **CON** (*n* allocated = 23; *n* completed outcome = 18): Standard of care	See Solheim et al. [71]	Visceral adipose tissue (cm^2^) (CT)	Within-group differences: **EX + NU + O**: PRE: 108.4 ± 67.6 vs. POST: 108.8 ± 66.1 **CON**: PRE: 99.9 ± 65.2 vs. POST: 94.9 ± 55.9 Between-group differences: **EX + NU + O vs. CON**: ES: 0.22
				Subcutaneous adipose tissue (cm^2^) (CT)	Within-group differences: **EX + NU + O**: PRE: 182.3 ± 114.5 vs. POST: 176.4 ± 108.5 **CON**: PRE: 160.6 ± 70.7 vs. POST: 149.4 ± 64.5 Between-group differences: **EX + NU + O vs. CON**: ES: 0.15
				Ratio VAT:SAT	Within-group differences: **EX + NU + O**: PRE: 0.7 ± 0.6 vs. POST: 0.7 ± 0.5 **CON**: PRE: 0.7 ± 0.5 vs. POST: 0.7 ± 0.4 Between-group differences: **EX + NU + O vs. CON**: ES: 0.25
				Total adipose area (cm^2^) (CT)	Within-group differences: **EX + NU + O**: PRE: 290.7 ± 154.0 vs. POST: 285.2 ± 149.5 **CON**: PRE: 260.5 ± 99.9 vs. POST: 244.3 ± 93.7 Between-group differences: **EX + NU + O vs. CON**: ES: 0.21
				Total adipose index (cm^2^/m^2^) (CT)	Within-group differences: **EX + NU + O**: PRE: 99.5 ± 52.7 vs. POST: 97.4 ± 51.2 **CON**: PRE: 93.3 ± 36.5 vs. POST: 87.4 ± 34.2 Between-group differences: **EX + NU + O vs. CON**: ES: 0.21
				Skeletal muscle mass index (cm^2^/m^2^) (CT)	Within-group differences: **EX + NU + O**: PRE: 45.9 ± 8.9 vs. POST: 45.0 ± 9.2 **CON**: PRE: 45.7 ± 8.6 vs. POST: 43.9 ± 9.4 ¥↓ Between-group differences: **EX + NU + O vs. CON**: ES: 0.26
Sheean et al. [80] **United States of America** RCT	**Population:** Metastatic breast cancer (*n* = 35) **Stage**: Metastatic: 100%	**EX + NU + O** (*n* allocated = 17; *n* complete outcome = 17): Multimodal (Supervised aerobic and resistance exercise, nutrition counseling) **CON** (*n* allocated = 18; *n* complete outcome = 18): Usual care waitlist control given monthly reminder of upcoming intervention.	12-week intervention **Exercise component (FITT):** **F**: 4 days/week **I**: moderate intensity **Time**: 150 min/week **Type**: AT = patient choice; RT = resistance bands. **Nutrition component:** Weekly phone calls, encouraging consumption of whole grains and 5+ fruits and vegetables daily. **Other component:** Rooted in social cognitive theory. **Adherence**: EX: attended 93% for supervised sessions NU + O: 84% for telephone sessions.	Appendicular skeletal muscle index (kg/m^2^) (DXA)	Within-group differences: **EX + NU + O**: MD: − 0.1 ± 0.4 **CON**: MD: 0.0 ± 0.2 Between-group differences: **EX + un + O vs. CON**: MD: − 0.0 ± 0.3
				Lean mass (kg) (DXA)	Within-group differences: **EX + NU + O**: MD: − 0.5 ± 1.6 **CON**: MD: − 0.3 ± 1.4 Between-group differences: **EX + NU + O vs. CON**: MD: − 0.4 ± 1.5
				Fat mass (kg) (DXA)	Within-group differences: **EX + NU + O**: MD: 0.3 ± 1.7 **CON**: MD: 0.3 ± 2.0 Between-group differences: **EX + NU + O vs. CON**: MD: 0.3 ± 1.8
				Body fat percent (%) (DXA)	Within-group differences: **EX + NU + O**: MD: 0.5 ± 1.3 **CON**: MD: 0.3 ± 1.2 Between-group differences: **EX + NU + O vs. CON**: MD: 0.4 ± 1.2
				Visceral fat mass (g) (DXA)	Within-group differences: **EX + NU + O**: MD: − 99 ± 181 ¥↓ **CON**: MD: − 81 ± 162 ¥↓ Between-group differences: **EX + NU + O vs. CON**: MD: − 89 ± 168 ¥↓
Mikkelsen et al. [79] **Denmark** RCT	**Population:** Pancreatic cancer, biliary tract cancer, small cell lung cancer (*n* = 84) **Stage**: Locally advanced: 14.3% Metastatic: 85.7%	**EX + NU + O** (*n* allocated = 43; *n* complete outcome = 29): Multimodal intervention (exercise + protein + PA+ counselling) **CON** (*n* allocated = 41; *n* completed outcome = 34): Usual care.	12-week intervention **Exercise components (FITT):** **F**: 2 days/week **I**: 10–15 RM **Time**: 60 min (Volume: 10–15 reps, 2–3 sets) **Type**: Supervised group-based resistance training. Individualized home-based walking program controlled with a pedometer. **Nutrition component**: Post-exercise protein supplementation intake (12–18 g) 2 days/week. **Other components:** Nurse-led support and counselling (holistic assessment of function) **Adherence**: EX: attended 69% of supervised exercise sessions and 75% adherence to the walking program. NU: NR	Lean mass (kg) (DXA)	Within-group differences: **EX + NU + O**: PRE: 47.3 ± 8.1 vs. POST: 48.7 ± 9.1¥↑ **CON**: PRE: 47 ± 9.2 vs. POST: 46.4 ± 9.1 Between-group differences: **EX + NU + O vs. CON**: MD: 0.9 ± 0.4 ¥↑
				Fat mass (kg) (DXA)	Within-group differences: **EX + NU + O**: PRE: 20.8 ± 8.1 vs. POST: 21.6 ± 7.6 **CON**: PRE: 22.4 ± 9.4 vs. POST: 22.7 ± 10 Between-group differences: **EX + NU + O vs. CON**: MD: 0.2 ± 0.6
				Lean mass (kg) (BIA)	Within-group differences: **EX + NU + O**: PRE:44.1 ± 8.5 vs. POST: 44.4 ± 9.6 **CON**: PRE: 42.9 ± 10.5 vs. POST: 41.9 ± 8.8 Between-group differences: **EX + NU + O vs. CON**: MD: −0.9 ± 1.3
				Fat mass (kg) (BIA)	Within-group differences: **EX + NU + O**: PRE:17.2 ± 8.8 vs. POST: 17.4 ± 8.5 **CON**: PRE: 18.5 ± 10.2 vs. POST: 18.9 ± 11.1 Between-group differences: **EX + NU + O vs. CON**: MD: 1.0 kg ± 1.0
Kim et al. [48] **Australia** RCT	**Population:** Prostate cancer (*n* = 40) **Stage**: Metastatic: 100%	**EX + O** (*n* allocated = 20; *n* complete outcome =13): Supervised aerobic and resistance training with psychological support **CON** (*n* allocated = 20; *n* complete outcome = 12): self-directed exercise	6-month intervention **Exercise component (FITT):** **F**: 3 days/week **I**: RT: 6–12 RM, HITT: RPE 8 AT: RPE 6 **Time**: RT: 2–5 sets, 6 exercises. HITT 3–6 bouts of 30–60 s. AT: 10–40 min. Progressive increase in time and volume. **Type**: Supervised RT and HITT 2 days per week and continuous cycling AT 1 day per week. **Other component** Psychological support **Adherence**: EX + O: attended 82.5 ± 13.0% of supervised exercise sessions.	Lean mass (kg) (DXA)	Within-group differences: **EX + O**: PRE: 53.1 ± 10.4 vs. POST: 50.6 (95%CI: 49.4 to 51.9) **CON**: PRE: 49.1 ± 8.2 vs. POST 50.7 (95%CI 49.4 to 51.9) Between-group differences: **EX + O vs. CON**: MD: NR
				Lean mass (%) (DXA)	Within-group differences: **EX + O**: PRE: 57.0 ± 3.9 vs. POST: 58.4 (57.1 to 59.6) **CON**: PRE: 59.8 ± 4.0 vs. POST: 57.7 (56.4 to 59) Between-group differences: **EX + O vs. CON**: MD: NR
				Lean mass index (kg/m^2^) (DXA)	Within-group differences: **EX + O**: PRE: 17.6 ± 1.9 vs. POST: 17.2 (16.8 to 17.4) **CON**: PRE: 16.7 ± 2.1 vs. POST: 17.0 (16.6 to 17.4) Between-group differences: **EX + O vs. CON**: MD: NR
				Fat mass (kg) (DXA)	Within-group differences: **EX + O**: PRE: 33.4 ± 10.5 vs. POST: 29.8 (27.9 to 31.8) **CON**: PRE: 26.9 ± 6.7 vs. POST: 32.1 (30.0 to 34.1) Between-group differences: **EX + O vs. CON**: MD: NR
				Body fat percent (%) (DXA)	Within-group differences: **EX + O**: PRE: 37.1 ± 4.4 vs. POST: 35.9 (34.4 to 37.5) **CON**: PRE: 34.4 ± 4.7 vs. POST: 36.7 (35.1 to 38.2) Between-group differences: **EX + O vs. CON**: MD: NR
Allen et al. [81] **United Kingdom** RCT	**Population:** Locally advanced esophagogastric cancer patients (*n* = 54) **Stage**: T1 = 1(2) T2 = 12 (22) T3 = 38 (70) T4 = 3 (6) N0 = 18 (33) N1 = 17 (31) N2 = 16 (30) N3 = 3 (6)	**EX + NU + O***n* allocated = 26; *n* complete outcome = 24: Prehabilitation Multimodal intervention (exercise + nutrition + psychological support) **CON** (*n* allocated = 28; *n* complete outcome = 28): Usual care with encouragement to get fitter during treatment.	15-week intervention **Exercise component (FITT):** **F**: Supervised in-clinic 2 days/week + Home-based 3 days/week **I**: AT: 40–60 HRR or 11–14 RPE and RT: 12–14 RPE **Time**: 60 min (Volume: 12 reps, 2 sets) **Type**: Prehabilitation supervised in clinic and unsupervised home-based AT and RT and flexibility. **Nutrition component:** Needs-based nutritional intervention with frequent, tailored, dietetic input from dieticians. **Other component** Psychological support: 6 face-to-face sessions with discussion of health status, strengths, recognition, resilience, or goal setting. **Adherence**: EX: attended 76 ± 14% of supervised exercise sessions and 65 ± 27% of home-based.NU + O: NR	Skeletal muscle index (cm^2^/m^2^) (CT)	Within-group differences: **EX + NU + O**: MD: −11.6 (95%CI –14.2 to –9.0) **CON**: MD: −15.6 (95%CI –18.7 to –15.4) Between-group differences: **EX + NU + O vs. CON**: MD: NR¥↑

¥: statistically significant change. ↑: increase. ↓: decrease. Abbreviations (in order of appearance): RCT, randomized clinical trial; *n*, sample of participants; EX, exercise intervention group; CON, control group; FITT, exercise frequency, intensity, time, and type; RT, resistance training; AT, aerobic training; NR, not reported; RM, repetition maximum; Min, minutes; Reps, repetitions; Kg, kilogram; DXA, Dual-energy X-ray absorptiometry; SD, standard deviation; 95%CI, 95% confidence interval; PRE, pre-intervention; POST, post-intervention; MD, mean difference; HR, heart rate; BIA, bioelectrical impedance analysis; RPE, rating of perceived exertion; NU, nutrition; EX + NU, exercise and nutrition intervention group; METs, metabolic equivalent of task; PA, physical activity; G, gram; G/KG, grams consumed per kilograms of body weight; Non-RCT, non-randomized controlled trial; Hz, hertz; μs, microsecond; S, seconds; kcal/kg/day, kilocalories consumed per kilograms of body weight per day; T, time point; Cm, centimeters; CT, computed tomography; NU + PA, nutrition and physical activity intervention group; NSCLC, non-small cell lung cancer; mL, milliliters; O, other component; EX + NU + O, exercise/physical activity and nutrition intervention group, as well as other component; ES, effect size; VAT:SAT, ratio of visceral adipose tissue to subcutaneous adipose tissue; M, meter; HITT, high intensity interval training; HRR, heart rate reserve.

## Data Availability

Not applicable.

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
