# Peer review of "The Effect of Exercise and Nutritional Interventions on Body Composition in Patients with Advanced or Metastatic Cancer: A Systematic Review"

_nutrients, 2022, doi:10.3390/nu14102110_

Round 1
Reviewer 1 Report
Nutrients Diet and PA in Metastatic Cancer 5.6.2022
I appreciate the opportunity to review this manuscript. It is a very important, yet understudied topic area and highly relevant to the journal’s readership, especially those interested in oncology and lifestyle behaviors. Overall is it well-written, engaging, and truly informative. The following comments are offered to help strengthen or clarify the reporting:
Abstract
Lines 26-27: Needs more directionality and vision, informed by review findings. Why clinical treatment? How about relevant outcomes assessment (eg, to evaluate treatment tolerance or quality of survival?)
Introduction
*The first two paragraphs of the introduction need to be carefully edited to be sure the statements are accurate. Several examples are highlighted below.
Lines 33-39: This first paragraph reflects a bias on the writing, and it is not substantiated across all cancers. Many persons with advanced/metastatic cancers experience weight gain, specifically breast and prostate. They may experience the symptoms articulated here, but the reference for this sentence supports cachexia (ref 4) and the inference is that wasting occurs as a universal phenomenon. Please revise to be more objective and accurate.
Lines 47-49: An important feature of cachexia is weight loss. Please include in the description, as this clinical feature differentiates this syndrome from sarcopenia.
Lines 51-52: Please clarify this sentence. For example, reference 28 is a meta-analyses of 9 studies and only one looked a psoas change. The term “rapidly accelerated” is an incorrect, misleading assessment of this work, as it infers changes in this group over time.
Lines 54-58: Are these studies that examine changes over time or do these studies reflect body composition at the time of imaging/body composition assessment? Please clarify.
Lines 62-63: Again the predominance of data depicts the prevalence of sarcopenia, not the incidence of sarcopenia. Please clarify this sentence removing the phrase “changes.” It is misleading and the reference (ref 16) reflects a cross-sectional assessment of this condition.
Lines 86-98: Well written, nicely highlighting the need for and novelty of this review.
Methods
*Well written and articulated. Figure is clear and easy to follow.
Results
*Easy to follow and comprehensive.
-Consider adding some verbiage to describe how body composition was measured. Specifically, please include what methods were used. This is articulated on lines 274-281; however, some of this information might be more helpful sooner.
- Table 1 is very informative. Is there a rationale for how the studies are presented? If so please articulate this on following the information on lines 204-205. It appears this may be stated on lines 220-224, but it is unclear and perhaps too late.
Discussion
*Very intriguing and insightful.
Lines 457-460: What is meant by false positives here? Can the authors better articulate their point? There are definitely concerns worth noting regarding the validity of these measures in clinical populations, specifically BIA. These tools were largely developed for healthy populations and yet they are used in clinical populations where “steady states” (fluid status specifically) may interfere with precision.
Conclusions
Line 484: “first-of-a-kind” – is this necessary, especially given the text on 354? It’s rather boastful and the novelty and originality of this review is well established in the introduction.
Line 492: Interrogate??
*Please consider adding something about the value of evaluating changes in body composition to improve relevant outcomes- quality of survivorship, treatment tolerance. In other words, why does body composition matter?
Reviewer 2 Report
Reviewer comments
Thank you for granting me the opportunity to review this interesting and topical piece of work. In this work, Barnes et al. conducted a systematic review of literature on the effect of exercise and nutritional interventions on body composition parameters (lean and fat mass) in patients with advanced metastatic cancer. Kindly, find below my comments for your response.
Abstract
The abstract is well-written. However, in the findings, because this research is on cancer, it will be expedient to indicate which common cancer type the findings related to. The authors could consider reporting the common cancer types that were reviewed.
Keywords: The authors stated in the Abstract that, the effect of the intervention on the “lean mass” and “fat mass” of the participants was their primary outcome. Consequently, I suggest that, the authors consider adding “lean mass” and “muscle mass” to the list of keywords. This is importantly so because the “body composition” is already captured in the “Title”.
Results:
In Figure 1, the authors should please indicate the criteria used for the article exclusion. For example, was it based on screening using article “Titles” and “Abstract”?
In Line 174, the authors use “16” but in Line 175, they use “one”. For consistency sake, the authors could stick to the use of either the “figures” or the “words”.
Line 178-180: In regards to the geographical location of the study areas, the authors should indicate the number of papers (n) prior to citing the references. That way, we can know the country with the highest number of studies which could be discussed.
Table 1. The authors should kindly revise Table one. There is lack of clarity on the authors use of “Pre vs Post training” in one of the columns. I can understand that the authors mean to indicate the effect of “Pre vs Post training” outcomes on the body composition parameters. This systematic review looks at the effect of intervention on two essential body composition parameters which include lean and fat mass. Consequently, the authors should indicate if there was an increase in the lean and fat mass. They could indicate that using “↑ or ↓” using the values from the between-group differences. Table 1. The geographical location of where the studies were conducted which the authors highlighted in the preceding section is one important aspect that can be captured in the discussion. It will therefore be interesting and relevant if the authors indicated in Table 1 “Countries where studies were conducted”.
Line 233: The authors use of “………..an additional component” should be explained as it may appear vague to the reader in its present state
Line 283: The authors should please consider revising the sentence
General comment: This manuscript is very well written. Great job to the authors.
